# UvrD helicase–RNA polymerase interactions are governed by UvrD's carboxy-terminal Tudor domain

Ashish A. Kawale [1,2] & Björn M. Burmann [1,2✉]

All living organisms have to cope with the constant threat of genome damage by UV light and other toxic reagents. To maintain the integrity of their genomes, organisms developed a variety of DNA repair pathways. One of these, the Transcription Coupled DNA-Repair (TCR) pathway, is triggered by stalled RNA Polymerase (RNAP) complexes at DNA damage sites on actively transcribed genes. A recently elucidated bacterial TCR pathway employs the UvrD helicase pulling back stalled RNAP complexes from the damage, stimulating recruitment of the DNA-repair machinery. However, structural and functional aspects of UvrD's interaction with RNA Polymerase remain elusive. Here we used advanced solution NMR spectroscopy to investigate UvrD's role within the TCR, identifying that the carboxy-terminal region of the UvrD helicase facilitates RNAP interactions by adopting a Tudor-domain like fold. Subsequently, we functionally analyzed this domain, identifying it as a crucial component for the UvrD–RNAP interaction besides having nucleic-acid affinity.

[1] Wallenberg Centre for Molecular and Translational Medicine, University of Gothenburg, Gothenburg 40530, Sweden. [2] Department of Chemistry and Molecular Biology, University of Gothenburg, Gothenburg 40530, Sweden. ✉email: bjorn.marcus.burmann@gu.se

  

Evolutionary conserved Transcription Coupled DNA-repair (TCR) pathways are achieved by RNA Polymerase (RNAP) acting as a global sensor for DNA lesions on actively transcribed genes, resulting in arrested transcription complexes[1]. As the stalled RNAP occludes the DNA lesions, therefore preventing a direct approach of the DNA-repair machinery (UvrAB complex in bacteria), the concerted action of a plethora of auxiliary factors is required to remove the arrested RNAP from the DNA lesion sites and to subsequently recruit the repair machinery. Mfd (also called Transcription-Repair Coupling Factor, TRCF) is a multi-domain bacterial protein for a long time believed to be the sole protein facilitating the coupling of transcription and DNA-repair machinery[2,3]. Mfd binds upstream of the stalled RNAP, inducing a forward translocation of the arrested RNAP by ATP-dependent 5′–3′ DNA translocase activity, leading to complete disassembly of the transcription machinery[4]. Via a direct interaction with UvrA, Mfd recruits the basic components of the DNA repair machinery[2]. Paradoxically, knockout of the *mfd* gene shows only minimal effects on the UV-sensitivity of bacterial cells, pointing to alternate TCR pathway(s)[5,6]. Intriguingly, recent studies propose alternate or additional TCR pathways mediated by the transcription factor NusA and the helicase UvrD[7–9]. UvrD, also termed Helicase II, binds directly to RNAP and is proposed to function within the TCR by using its inherent ATPase activity for backtracking the stalled RNAP without displacing it altogether[8]. This elegant mechanism, facilitated by UvrD, would ensure the possibility of transcription restart from the damage site once the DNA repair is done, circumventing the need for Mfd-dependent abortive transcription[10]. Although this mode of action of UvrD within the TCR proposed by Nudler and coworkers is still under intense debate[9,11,12], UvrD's binding to RNAP and its ability to induce RNAP backtracking are nevertheless widely accepted[5,13].

UvrD helicase is a multi-domain DNA helicase with a size of 82 kDa[14]. Biophysical characterization indicates that ATP-dependent DNA translocation, as well as helicase activity, are regulated by switching between monomeric and dimeric forms of UvrD[15]. Crystallographic studies on *E. coli* UvrD revealed the presence of four distinct domains namely, 1A and 2A, harboring the ATP binding site, required for ATP-hydrolysis; whilst 1B and 2B are involved in DNA binding[16]. Conformational dynamics governed by a 2B domain rotation in the absence and presence of DNA are supposed to be important drivers for its helicase as well as its DNA sliding activities[17,18].

Extensive biochemical studies have proposed that the extreme carboxy-terminal region of the UvrD helicase acts as a protein hub for a large variety of interaction partners (e.g., ssDNA, UvrB, MutL), enabling UvrD to perform its diverse cellular roles in biological processes such as mismatch repair, nucleotide excision repair, and recombination[19–22]. Recent studies also propose that this carboxy-terminal region might play an important role in UvrD's interaction with RNAP[23].

Despite being such an important part of the protein, the structure-function relationship of the carboxy-terminal extension of UvrD remains so far highly ambiguous. It was hypothesized that this region is mostly unstructured rendering polydispersity to the full-length protein in vitro[16,21]. Hence, this region was withdrawn for the structural studies on the UvrD helicase from *E. coli*[16,17], *D. radiodurans*[24], and the related *G. stearothermophilus* PcrA[25]. Intriguingly, latest studies on *G. stearothermophilus* PcrA revealed that its carboxy-terminal region adopts a Tudor-domain-like fold through facilitating RNAP interactions[23,26]. Based on the high sequence similarity between PcrA and UvrD, the authors also suggest the presence of a Tudor-domain fold in the UvrD carboxy-terminal region with similar properties[26].

In light of the recently discovered UvrD's role in TCR pathway, we set out to study the structural basis of the UvrD-RNAP interaction, characterizing the structure and functions of the UvrD carboxy-terminal region. Using advanced biomolecular NMR spectroscopy, we are for the first time able to study the protein at atomic resolution in solution showing that the *E. coli* UvrD carboxy-terminal region is adopting a stable fold consisting of five strongly bent antiparallel β–strands resulting in a Tudor-domain-like fold. Interaction studies by NMR spectroscopy and Biolayer Interferometry (BLI) revealed the importance of this domain for the RNAP interaction of UvrD as well as an inherent affinity to single and double-stranded DNA. Furthermore, structural and functional comparison with the corresponding RNAP interaction domain of Mfd showed functional divergences to UvrD and displacement experiments revealed that both proteins exploit non-overlapping types of RNAP interaction surfaces despite being unable to bind RNAP simultaneously.

## Results

**Solution NMR spectroscopy reveals a Tudor-domain like fold at the extreme carboxy-terminal region of UvrD.** Based on multiple sequence alignments with PcrA (*S. aureus*) and Rep helicase (*E. coli*) combined with available structural data from crystallographic studies[16,17], we defined the *E. coli* UvrD carboxy-terminal region, which will be termed throughout as the UvrD-CTD (encompassing residues 645–720) (Fig. 1a and Supplementary Fig. 1). This domain boundary was chosen as this region was missing from previously reported UvrD structures[16,17] and was derived from the structural studies observed for the *G. stearothermophilus* PcrA-CTD[26]. Initially, we performed size–exclusion chromatography coupled with multi-angle light scattering (SEC-MALS) and an NMR estimation of the rotational correlation time to assess the oligomeric state of the construct. Under the chosen experimental conditions, SEC-MALS data shows that the UvrD-CTD is a monomer in solution with an apparent molecular weight of $7.8 \pm 0.5$ kDa, closely matching the theoretical value of 8.4 kDa (Fig. 1b). Although the monomeric state is also reflected by the estimation of the effective rotational correlation time ($\tau_c$) with 3.63 ns, the obtained value is about 30% lower than expected for a protein of this size (Fig. 1c). This smaller than expected $\tau_c$ value is indicative of the presence of a highly flexible polypeptide segment dominating the analysis in the used approach[27].

The 2D [$^{15}$N,$^1$H]-NMR spectrum of [$U$-$^{15}$N]-UvrD-CTD ($U$ denotes uniform labeling) yielded a well-dispersed high-quality spectrum indicating the presence of stable secondary structure elements. Importantly, the observed chemical shifts of the 2D [$^{15}$N,$^1$H]-NMR spectrum of the UvrD-CTD construct match remarkably well with a sub-set of resonances of the full-length [$U$-$^2$H,$^{15}$N]-UvrD in a 2D [$^{15}$N,$^1$H]-NMR spectrum (Fig. 1d), indicating that the UvrD-CTD fold is well-preserved within the full-length protein. Moreover, analysis of the 2D [$^{15}$N,$^1$H]-NMR spectrum of the full-length proteins shows that the intensity of the resonances associated to the CTD is stronger compared to the other resonances, implying that the CTD moves rather freely compared to the rest of the UvrD-protein.

Almost complete sequence-specific backbone and side-chain resonance assignment of the UvrD-CTD (~95%) could be obtained by standard approaches (Supplementary Fig. 2). The secondary chemical shifts of the Cα and Cβ moieties revealed a strong propensity for a β-stranded structure comprising of five distinct β-strands with the first 29 amino-terminal residues being highly disordered (Fig. 1e). Based on 3D $^{15}$N-edited and $^{13}$C-edited NOESY spectral analysis several inter-strand long-range NOEs ascribed to five antiparallel β-strands could be identified (Supplementary Fig. 3a).

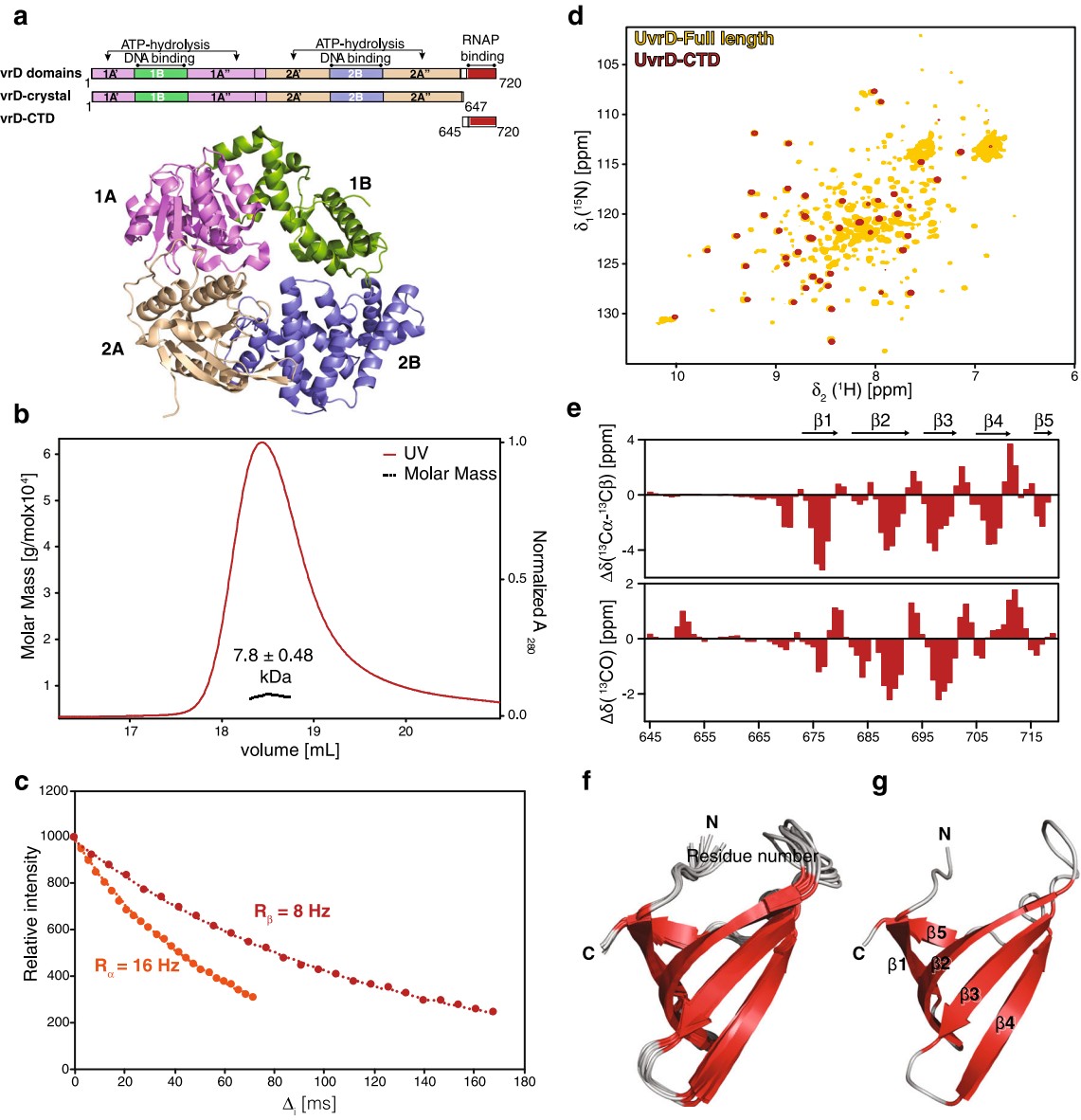

**Fig. 1 UvrD-CTD is a folded extension of UvrD. a** Schematic representation of UvrD domain architecture and constructs used in a recent crystallographic study and UvrD-CTD in this study and ribbon representation of the reported apo-UvrD crystal structure (PDB:3LFU), indicating the location of each domain. **b** SEC-MALS elution profile of UvrD-CTD recorded on 2 mg/ml protein in PBS buffer pH 7.4 at room temperature. **c** [$^{15}$N,$^{1}$H]–TRACT data for determining the rotational correlation time $\tau_c$. The 1D $^{1}$H signal intensity of UvrD-CTD was integrated and plotted against the relaxation period $T$. **d** Overlay of 2D [$^{15}$N,$^{1}$H]-NMR spectra of [$U$-$^{2}$H,$^{15}$N] labeled UvrD full-length and [$U$-$^{15}$N] labeled UvrD-CTD construct acquired in NMR buffer at 310 K. **e** Derived secondary structure elements of UvrD-CTD based on combined $^{13}$Cα and $^{13}$Cβ, as well as $^{13}$C′ secondary chemical shifts, for the UvrD-CTD construct indicating the presence of five β-strands within the extreme carboxy-terminal region of UvrD. **f** Ribbon representation of the ensemble of 10 lowest energy solution NMR structures of UvrD-CTD after water refinement showing a characteristic Tudor-domain-like fold (five highly bent antiparallel β-strands (β1–β5)). β-strands and connecting loops are indicated in red and gray, respectively. The amino-terminal non-converging residues 645–668 residues are excluded from the representation. **g** Cartoon representation of UvrD-CTD with secondary structure elements and termini indicated.

Solution NMR structure calculation of the CTD resulted in a well-converging structural bundle as represented by an ensemble of 10 structures (Fig. 1f) yielding an RMSD of 0.44 Å for the backbone atoms of the stable secondary structure elements (673–720) (Table 1). The core structural elements comprise of five strongly bent antiparallel β–strands adopting a β-barrel-type Tudor-domain-like fold, with residues 673–678, 681–690, 697–702, 705–711, 717–719 forming the β1, β2, β3, β4, and β5 strands, respectively (Fig. 1g). We did not observe any apparent NOEs for the short α-helical turn indicated by the secondary chemical shifts comprising of residues 712–716 located

in between the β4 and β5 strands, suggesting its transient nature. The highly flexible amino-terminus did not converge in the structure calculation owing the lack of medium- and long-range NOEs. This observation is in line with a flexible attachment of the CTD to the rest of the UvrD-protein, as initially evidenced by the overlay of the sub-spectra (Fig. 1d). Analysis of the electrostatic surface potential revealed a distinct distribution of charges, where the majority of the front side shows the presence of weak charges on the surface. On the backside of the protein structure positively charged residues are clustered (Supplementary Fig. 3b), suggesting that potentially this side of the UvrD-CTD might be critical

**Table 1 Structural statistics of solution NMR structures of UvrD-CTD and Mfd-RID.**

|  | UvrD-CTD | Mfd-RID |
|---|---|---|
| **NMR distance and dihedral constraints** | | |
| Distance constraints | | |
| Total NOE | 2205 | 2606 |
| Sequential ($\lvert i - j\rvert \leq 1$) | 1446 | 1698 |
| Medium-range ($\lvert i - j\rvert < 5$) | 153 | 193 |
| Long-range ($\lvert i - j\rvert \geq 5$) | 606 | 715 |
| Total dihedral angle restraints | | |
| $\phi$ | 41 | 52 |
| $\psi$ | 41 | 55 |
| **Structure statistics\*** | | |
| Violations (mean ± s.d.) | | |
| Distance constraints (Å) | 0.014 ± 0.001 | 0.021 ± 0.0019 |
| Dihedral angle constraints (°) | 0.291 ± 0.075 | 0.623 ± 0.056 |
| Deviations from idealized geometry | | |
| Bond lengths (Å) | 0.003 ± 0.00008 | 0.004 ± 0.0001 |
| Bond angles (°) | 0.431 ± 0.014 | 0.496 ± 0.011 |
| Impropers (°) | 1.224 ± 0.067 | 1.101 ± 0.061 |
| Average pairwise r.m.s. deviation (Å) | | |
| Heavy | 0.91 | 0.71 |
| Backbone | 0.29 | 0.27 |

*For 10 lowest energy structures after water refinement.

for the function of this domain. It is important to note that this side of the UvrD-CTD structure also shows a cluster of four aromatic residues namely H678, F681, W709, and Y714 potentially forming an aromatic cage (Supplementary Fig. 3C), which is a key feature of Tudor domains mediating protein-protein interactions via recognizing methylated lysine-arginine residues[28,29].

**Inherent dynamic properties of UvrD-CTD.** We next evaluated the backbone dynamics of the UvrD-CTD over a broad range of timescales by performing backbone NMR relaxation measurements[30]. By measuring the steady-state heteronuclear $^{15}N\{^{1}H\}$-NOE (hetNOE) and $^{15}N$ longitudinal ($R_1$) relaxation rates we probed the pico- to nanosecond motions of the N–H bonds (Fig. 2a, b). Whereas high hetNOE values and low $R_1$ rates indicate rigid and stably folded regions the inverse, low hetNOE values as well as high $R_1$ rates point to flexible and unfolded segments. Consistent with the structural characterization, the hetNOE data indicated that the first 29 amino acids of the UvrD-CTD are highly flexible as evidenced by negative hetNOE values followed by the structured region comprising the Tudor domain fold. The average hetNOE value for the residues comprising sheets β1–β4 was 0.65 indicating a rather stable fold, whereas the values for β5 were reduced to 0.5 indicating more extensive flexibility on the fast timescale for this strand. Nevertheless, the obtained values for the CTD, in general, are well below the theoretical maximum expected at 16.4 T (700 MHz $^{1}H$ frequency) of 0.86, indicating the presence of a large amplitude of global fast motions within the whole domain. Interestingly, the aromatic residues of the potential aromatic cage as well as adjacent F702 are in regions showing increased motions on the pico- to nanosecond timescale, possibly indicating dynamic adaptions for interaction with a variety of binding partners (Fig. 2c). The obtained average $R_1$ rate for the folded β-stranded region is determined to be 1.81 s$^{-1}$, in line with the magnitude of the obtained hetNOE values.

Likewise, we measured the transverse ($R_2$) relaxation rates to detect slow-time scale motions in the range of micro- to milliseconds. The average calculated $R_2$ rate derived from $R_{1\rho}$ for the folded β-stranded region was 5.60 s$^{-1}$, whereas for the amino terminus exhibited 5.92 s$^{-1}$. This flat profile indicates that we could not detect exchange contributions on the low μs-timescale, as the used 2000 Hz radiofrequency field would refocus dynamic contributions of the $^{15}N$ $R_2$ rate that are slower than 80 μs (Supplementary Fig. 4a). We compared this data with the profile obtained for the slow relaxing $^{15}N[^{1}H]$ doublet component ($R_{2\beta}$) reporting on contributions in the higher μs–ms timescale, clearly showing increased dynamics for the amino-terminus and the loops (Supplementary Fig. 4b). Together, these $R_1$ and $R_2$ correspond to an average rotational correlation time ($\tau_c$) of 3.64 ns for the structured region (Supplementary Fig. 4c), similar to previous values determined for isolated Tudor domains[31].

We next analyzed the obtained $^{15}N$ relaxation parameters using the Lipari–Szabo model-free approach[32,33]. The obtained $S^2$ values were in the range of 0.5–0.85, where higher $S^2$ values were exhibited by the β strands, although not reaching the maximal value of 1, indicating the inherent flexibility of the protein domain which is also evidenced by the obtained chemical exchange rates ($R_{ex}$) (Fig. 2d, e). Remarkably, besides the loops connecting the β-sheets also parts of the aromatic cage show elevated exchange contributions on the μs–ms timescale (F702, W709; Fig. 2f).

**Corresponding Mfd-RID also encompasses a Tudor domain fold in solution.** To compare the properties of UvrD with Mfd in solution, we delineated the Mfd RNAP binding region (472–547; comprising Domain 4), termed Mfd-RID (RNAP Interaction Domain), based on the full-length Mfd crystal structure (PDB ID:2EYQ)[34] (Fig. 3a). SEC-MALS indicated the presence of a monomeric protein form with an apparent molecular weight of 8.2 ± 0.16 kDa (Fig. 3b) consistent with the determined rotational correlation time of 5.6 ns (Fig. 3c). The 2D [$^{15}N,^{1}H$]-NMR spectrum recorded for Mfd-RID yielded well-dispersed resonances, facilitating almost complete sequence-specific resonance assignment (~94%; Supplementary Fig. 5a–d). The chemical shift derived secondary structure elements indicated the presence of five β-strands consistent with the domain 4 structure deduced from the reported crystal structure[34]. Furthermore, we observed propensity for an additional β-strand formed by the carboxy-terminal residues of the construct (537–545), comprising a linker region in the full-length crystal structure (Fig. 3d). Based on the extent of the secondary chemical shifts we estimate that this additional β-strand is partially stable and only populated to about 30% (Fig. 3d). Using a truncated Mfd-RID construct (472–533), lacking this carboxy-terminal linker region, renders the protein domain unfolded in solution, suggesting that this region is essential for stabilizing the Tudor domain fold of Mfd-RID (Supplementary Fig. 5e). Based on $^{15}N$-edited and $^{13}C$-edited 3D NOESY spectra analysis we could identify the characteristic NOE pattern for antiparallel β-strands (Supplementary Fig. 6a).

The NMR structure of Mfd-RID resulted in a well-converged structural ensemble confirming the presence of a compact core comprising of five strongly bent antiparallel β-strands adopting β-barrel-like fold (Fig. 3e). The residues 480–485, 488–500, 504–514, 516–522, 529–531 form β1, β2, β3, β4, and β5 strands, respectively (Fig. 3f). Although we did not observe any apparent backbone amide NOEs for the residues comprising β6 strand, structure calculation resulted in a well-converged loop indicating β-sheet structural propensity as already suggested by the secondary chemical shifts. These structural elements of Mfd-

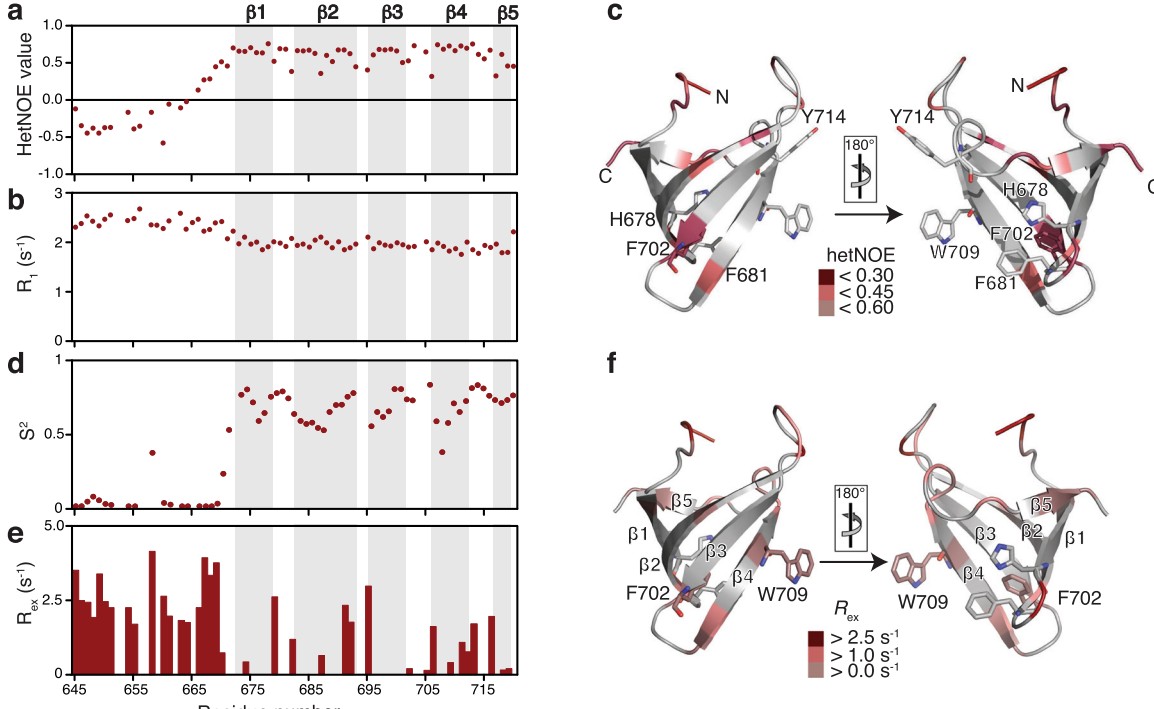

**Fig. 2 Inherent backbone dynamics of UvrD-CTD. a**, **b** Fast-timescale (pico- to nanosecond) motions of UvrD-CTD assessed by measuring **a** $^{15}$N{$^{1}$H}-NOE values and **b** $^{15}$N $R_1$ relaxation rates. **c** Residues showing decreased hetNOE values and therefore experiencing large amplitude of fast-scale motions are highlighted in the red gradient as indicated. Key aromatic residues are shown as sticks. **d**, **e** $^{15}$N-NMR relaxation parameters were analyzed using the Lipari–Szabo model-free formalism[30,31] to determine the generalized order parameters $S^2$ (**d**) and the chemical exchange rates $R_{ex}$ (**e**). The order parameter $S^2$ reflects on the motions in ps–ns timescales whereas chemical exchange rates $R_{ex}$ indicate motions on µs–ms timescales. **f** Chemical exchange contribution plotted on the UvrD-CTD structure for residues experiencing $R_{ex}$ as indicated.

RID match very well with the previously reported crystal structure of full-length protein with an RMSD of 0.83 Å (Supplementary Fig. 6b). Electrostatic surface potential analysis revealed a random distribution of weak charges on the surface of the Mfd-RID (Supplementary Fig. 6c). Importantly, although Mfd-RID contains five tyrosine residues, their distribution on the structure does not indicate the formation of a characteristic aromatic cage (Supplementary Fig. 6d). Thus, our solution NMR data corroborate that the isolated RID retains a partial Tudor domain-like fold lacking the characteristic aromatic residues crucial for mediating protein-protein interactions.

**Relaxation properties of Mfd-RID.** Assessing the dynamical properties of Mfd-RID with the same approach as outlined above confirmed the secondary structural elements of the RID with consistent relaxation rates across the β–strands (Supplementary Fig. 7). For the structured part of the protein, the average hetNOE value observed for the β-sheets β1–β5 was 0.81 close to the theoretical maximum value of 0.86, indicating a stable fold devoid of large amplitude ps–ns motions (Supplementary Fig. 7a). In agreement with its transient nature, the average hetNOE value for β6 strand was slightly reduced to 0.69. The average relaxation rates $R_1$ *and* $R_2$ were determined to be 1.70 and 5.81 s$^{-1}$, respectively (Supplementary Fig. 7b–d). Residues mainly located in the loop regions and β6 strand along with a few residues belonging to the β2 and β3 strands show higher than average $R_2$ value, indicating their propensity for undergoing conformational exchange on a slower (µs–ms) timescale. The average rotational correlation time $\tau_c$ of Mfd-RID is 4.29 ns for the structured parts of the protein (Supplementary Fig. 7g). The obtained $S^2$ values using model-free analysis were in the range of 0.62–0.85, where higher $S^2$ values were exhibited by the β strands indicating a

compact protein fold and lower values were observed mainly for the connecting loops as well as the β6 strand indicative of their inherent flexibility. (Supplementary Fig. 7e). Similarly, chemical exchange rate ($R_{ex}$) analysis was fully consistent with the $R_2$ rates, where loop residues and the β6 strand, as well as parts of the β2 and β3 strands, exhibited conformational exchange on the µs–ms timescale (Supplementary Fig. 7f). In contrast to UvrD-CTD, the Mfd-RID did not show any implications for a large extent of inherent dynamics on the fast (ps–ns) as well as the slow (µs–ms) timescales, consistent with a stable protein domain not undergoing any structural adaptions.

**UvrD-CTD is important for the RNAP interaction.** To address the RNAP-binding properties of UvrD-CTD, we performed NMR titrations adding unlabeled RNAP core-enzyme (subunits α$_2$ββ'ω) to [$U$-$^{15}$N]-UvrD-CTD (Fig. 4a). Already, upon addition of 0.1 molar ratio of RNAP, the backbone amide resonances of UvrD-CTD exhibited severe line-broadening, due to the formation of a large UvrD-RNAP complex (>300 kDa; Fig. 4b). This effect was most pronounced for the structured part of the UvrD-CTD whereas amino-terminal residues experienced weaker line-broadening, highlighting that the Tudor domain-fold of UvrD-CTD mediates the RNAP interaction (Fig. 4c).

To assess the RNAP–UvrD-CTD interaction quantitatively we used Bio-Layer Interferometry (BLI) analysis to characterize the UvrD-CTD-RNAP interaction. The dissociation constant ($K_D$) between UvrD-CTD and RNAP core-enzyme was determined to be 77 ± 0.3 nM (Fig. 4d), whereas that of UvrD full-length and the RNAP core-enzyme was found to be 1.0 ± 0.13 µM (Fig. 4e and Supplementary Table 1), weaker than the binding affinity of the isolated UvrD-CTD towards RNAP possibly suggesting possible stearic hindrance imposed by the rest of the UvrD domains.

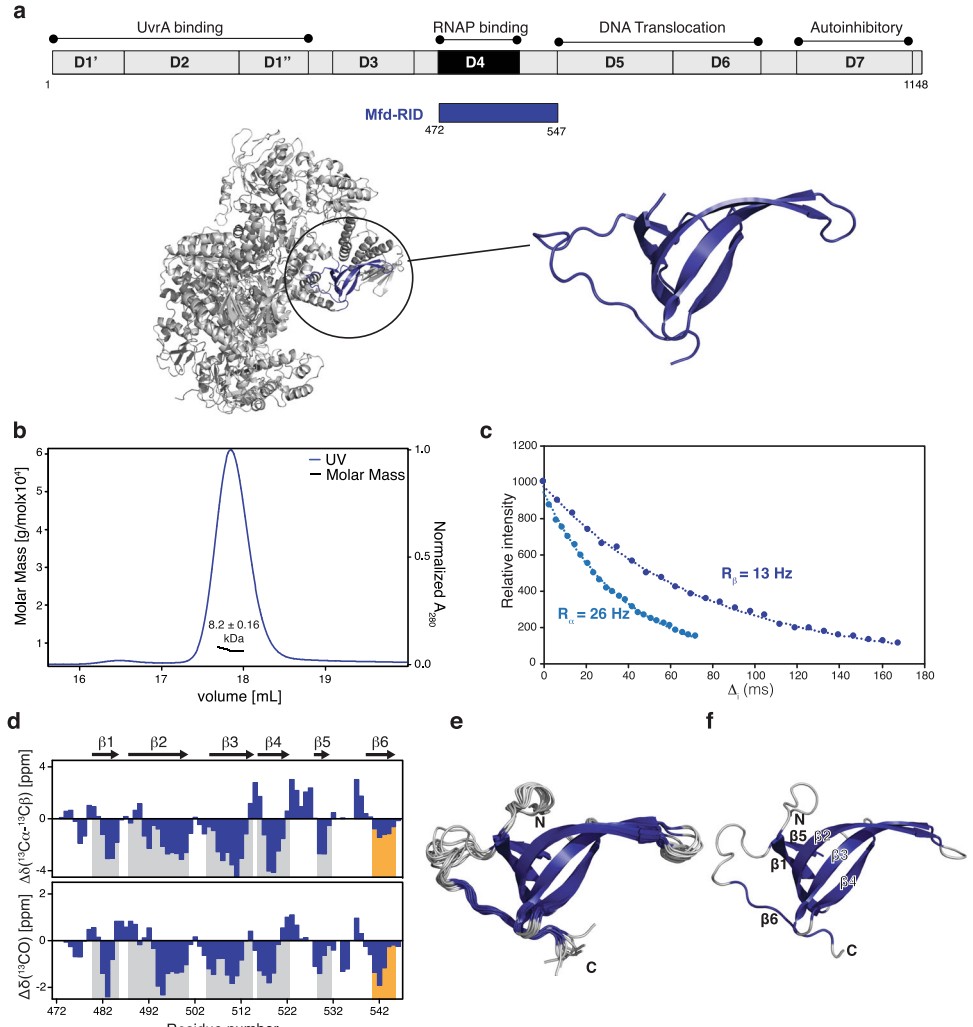

**Fig. 3 Solution structure of the isolated Mfd-RID. a** Schematic diagram of the Mfd domain arrangement and the Mfd-RID construct used in this study. Ribbon representation of the reported crystal structure (PDB:2EYQ) with position and structure of the RID domain shown in the inset. **b** SEC-MALS elution profile of Mfd-RID in PBS buffer, pH 7.4 at room temperature. **c** 1D [$^{15}$N,$^{1}$H] TRACT data for Mfd-RID. The 1D $^{1}$H signal intensity of UvrD-CTD was integrated plotted against the relaxation time $T$. **d** Secondary chemical shifts (combined $^{13}$Cα, $^{13}$Cβ and $^{13}$C') for Mfd-RID indicate the presence of six β-strands. **e** Ribbon representation of an ensemble of the 10 lowest energy solution NMR structures of Mfd-RID after water refinement showing Tudor-domain-like fold characterized by five highly bent antiparallel β-strands and one irregular β-strand (β6). β-strands are colored in blue whereas connecting loop are shown in gray. **f** Cartoon representation of Mfd-RID with secondary structural elements and termini indicated.

Although the possibility of large conformational changes in the UvrD domain arrangement upon RNAP binding cannot be ruled out. To check whether the deletion of CTD leads to a reduced RNAP-binding, confirming the importance of the CTD for the RNAP interaction, we performed the BLI assay with a truncated UvrD lacking the CTD (UvrDΔCTD). In complete agreement with our hypothesis, we observed a two-fold decrease in the RNAP binding affinity upon CTD deletion ($K_D = 1.90 \pm 0.15$ μM), highlighting the important contributions of the CTD to the overall UvrD–RNAP binding (Fig. 4f and Supplementary Table 1). The observed minute 2-fold change is consistent with previous reports, where despite UvrD-CTD deletion, UvrD could still perform RNAP backtracking activity, albeit to a reduced extent, indicating RNAP binding contributions from other protein parts[26]. This feature of UvrD-CTD points to an essential function as a protein-ligand binding hub facilitating the RNAP interaction.

**UvrD-CTD Tudor domain interacts with DNA non-specifically.** Recent studies have discerned the Tudor domain's capacity for mediating nucleic acid interactions[35,36], in addition to their initially described ability to recognize methylated lysine-arginine residues[29,37]. Moreover, previous biochemical studies have also highlighted the importance of the UvrD carboxy-terminal region (residues 618–720) by reporting the failure of UvrDΔ102C (deletion of 107 carboxy-terminal amino acids) replacing full-length UvrD in nucleotide excision repair and other UvrD functions[22]. Nevertheless, these results have to be treated with caution due to the fact that also parts of domain 2A were deleted, possibly affecting the structural integrity of this domain. Hence, in order to investigate the DNA binding properties of UvrD-CTD, we performed NMR titrations with a 17mer single-stranded DNA (ssDNA) and a self-complementary 35mer where the first 28 nucleotides form double-stranded DNA (dsDNA) with 3' overhangs comprising 7 nucleotides, used in structural studies before[16]. Upon addition of the 17mer, a distinct subset of CTD backbone amide resonances exhibited slight chemical shift changes accompanied by severe line-broadening, indicating the protein-DNA complex formation using a discrete set of CTD residues (Fig. 5a). Most of the affected residues are located on the

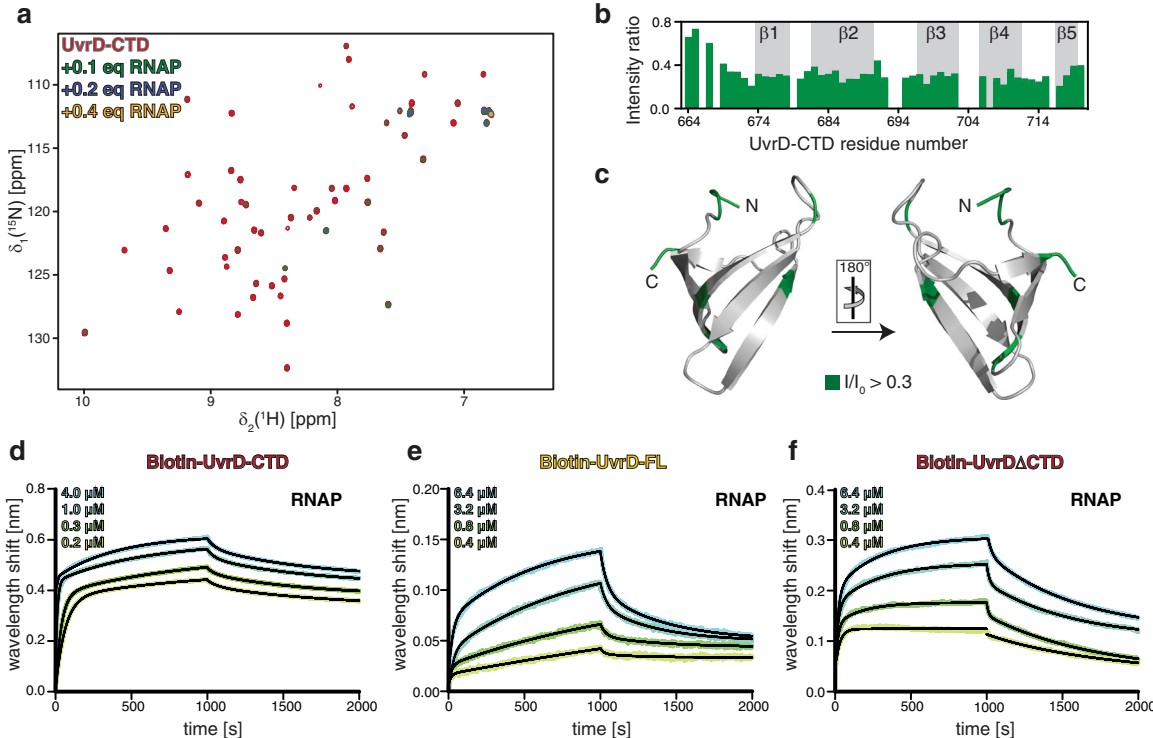

**Fig. 4 UvrD-CTD binds directly to RNAP. a** Overlay of 2D [$^{15}$N,$^{1}$H]-NMR spectra of [$U$-$^{15}$N] UvrD-CTD in the absence (red) and in the presence of increasing amounts of RNA polymerase (as indicated) acquired in the PBS buffer at 298 K. **b** The ratio of individual peak intensities in the presence of 0.1 molar excess of RNA polymerase (core enzyme) to the apo-UvrD-CTD, plotted against residue numbers. Residues 645–663 were removed from the analysis due to severe exchange broadening under the used conditions. **c** Effects of the RNAP interaction plotted on the UvrD-CTD structure as indicated. **d–f** Biolayer interferometry (BLI) data analysis of UvrD-CTD **d**, UvrD-FL. **e** UvrD-ΔCTD **f** binding to RNAP (core enzyme), respectively. Analyte concentrations are indicated in each figure. Non-linear least-square fits to the experimental data are indicated by the black lines.

loops connecting β-strands with the loop between the β4 and β5 strand together with the unstructured amino-terminal region (Fig. 5b). Upon addition of dsDNA (35mer), a similar effect in terms of chemical shift changes and line-broadening was observed (Fig. 5c), with the loops adjoining β-strands showing major changes along together with the amino-terminal unstructured region (Fig. 5d), indicating that the DNA binds to one side of the Tudor domain fold. In the presence of ssDNA, chemical shift changes were observed on three of the four key aromatic residues (F681, W709, Y714), pointing to base-stacking as the major binding contribution. On the other hand, chemical shift changes were only observed on an aromatic residue F681 in the presence of dsDNA. Importantly, the DNA binding surface of the UvrD-CTD overlaps almost perfectly with the positively charged residues on the UvrD-CTD surface (Supplementary Fig. 3b), highlighting the importance of these residues and the positioning of the neighboring aromatic residues in mediating DNA binding.

Next, we used the BLI assay to quantitate the binding affinity of CTD with the different DNA oligonucleotides. BLI data confirmed that the CTD binds to both ssDNA (17mer) and dsDNA oligonucleotides (35mer) whereas we observed a two-fold higher binding affinity for dsDNA (1.9 ± 0.17 µM) in comparison to the ssDNA (3.8 ± 0.47 µM) (Fig. 5e, f and Supplementary Table 2), suggesting that UvrD-CTD has a slight preference for the dsDNA over ssDNA, in agreement with the observations in the NMR titration experiments.

**Mfd-RID interacts with RNAP but not DNA.** Even though the role of Mfd in the TCR pathway is well established, including that domain 4 is critical for Mfd–RNAP interactions, the atomic-level details of this interaction remain partially elusive. The crystal

structure of the complex between *Thermus thermophilus* Mfd-RID$^{321–387}$ as well as *Thermus aquaticus* RNAP β1 domain from the β-subunit (discontinuous β1 domain comprising β1a$^{17–139}$ and β1b$^{334–395}$ connected by a –Gly–Gly– linker) highlight the importance of key residues for the RID-RNAP interaction[38]. Hence, in order to characterize the details of the Mfd-RID–RNAP interaction also in the *E. coli* TCR system, we determined the RNAP binding patch of Mfd-RID by NMR. Upon serial addition of protonated RNAP core enzyme, we observed an almost uniform decrease in the intensities of backbone amide resonances of the Mfd-RID in a similar manner observed for UvrD-CTD (Supplementary Fig. 8a, b). This global loss of signal intensity is likewise due to the large complex formed. Detailed analysis revealed effects almost all over the protein, with main changes on the backside of the protein composed of the amino-terminal and carboxy-terminal protein sections (Supplementary Fig. 8b).

The dissociation constant between Mfd-RID and RNAP core enzyme was determined to be 250 ± 52 nM using the BLI assay (Supplementary Fig. 8c, d), indicating that the Mfd-RID binds RNAP core enzyme about 3-fold weaker than UvrD-CTD. The binding affinity between Mfd-full length protein and RNAP core enzyme was found to be 310 ± 14 nM, which is about 3 times stronger than that of the UvrD-full length and RNAP (1 µM) (Supplementary Table 1). In contrast to UvrD, the observation of no alterations in the binding affinity towards RNAP indicate that the Mfd-RID is the sole RNAP interacting domain, in perfect agreement with previous structural and functional studies[34,38].

Next, we tested the DNA binding properties of Mfd-RID, by performing similar NMR titrations with the ssDNA and dsDNA as described for UvrD-CTD. Upon serial addition of ssDNA, we could not observe any notable chemical shift changes in Mfd-

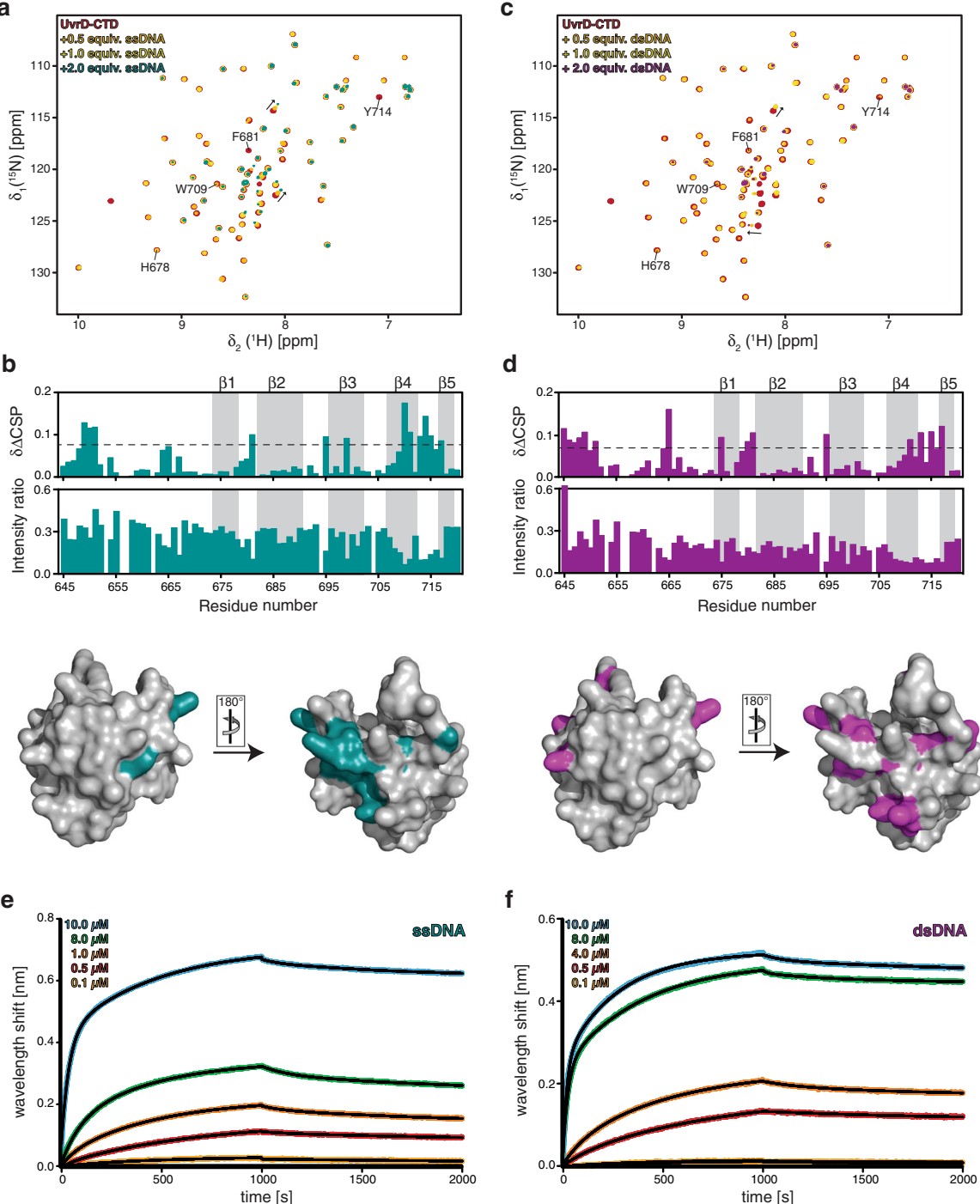

**Fig. 5 UvrD-CTD has a weak inherent DNA binding capacity. a** Overlay of 2D [$^{15}$N,$^1$H]-NMR spectra of [$U$-$^{15}$N] UvrD-CTD in the absence (red) and in the presence of increasing amounts of 17mer (ssDNA) as indicated. Residues involved in aromatic cage formation are annotated along with arrows indicating CSPs of few representative residues. **b** Chemical shift perturbation and intensity ratio plotted against the UvrD-CTD residue numbers in the presence of two molar equivalent of ssDNA. Secondary structural elements are highlighted with a gray background. Residues undergoing stronger CSPs than twice the standard deviation (S.D.) are mapped in turquoise on the surface representation of the UvrD-CTD shown in two different orientations. **c** Titration of UvrD-CTD with dsDNA shown by the overlay of 2D [$^{15}$N,$^1$H]-NMR spectra of [$U$-$^{15}$N] UvrD-CTD in the absence (red) and in the presence of increasing amounts of 35mer dsDNA. CSPs for the few representative residues are marked with arrows whereas residues involved in putative aromatic cage formation are highlighted. **d** Chemical shift changes and intensity changes upon dsDNA binding (two molar excess) plotted against residue numbers of UvrD-CTD, with positions of β-strands highlighted in the gray background. The surface representation of the UvrD-CTD marked with the residues undergoing stronger CSPs than twice the S.D. in magenta and shown in two different orientations. Both NMR titrations were performed in UvrD sample buffer at 298 K. **e**, **f** Kinetic analysis by BLI of biotinylated ssDNA (**e**) and biotinylated dsDNA (**f**) with varying UvrD-CTD concentration (indicated in each figure). Black lines represent non-linear least-square fits to the experimental data.

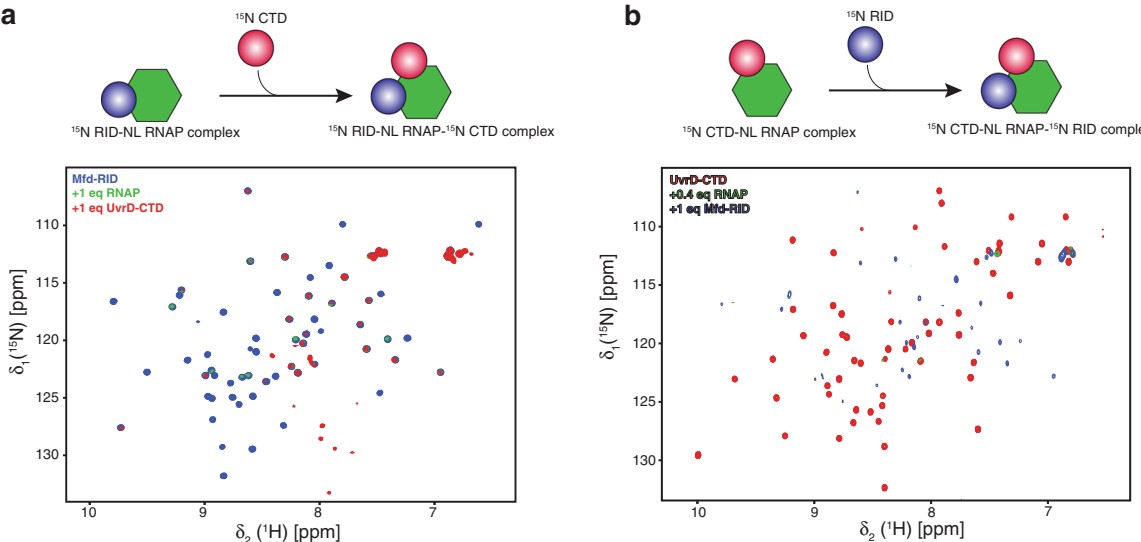

**Fig. 6 UvrD-CTD and Mfd-RID do not compete for RNAP binding. a** Schematic representation of experimental design as well as overlay of 2D [$^{15}$N,$^{1}$H]-NMR spectra of [$U$-$^{15}$N] Mfd-RID in the absence (blue) and in the presence (green) of one molar equivalent of RNAP and [$U$-$^{15}$N] UvrD-CTD (red), respectively. **b** Schematic representation of experimental setup as well as overlay of 2D [$^{15}$N,$^{1}$H]-NMR spectra of [$U$-$^{15}$N] UvrD-CTD in the absence (red) and in the presence (green) of 0.4 molar equivalents of RNAP and one molar equivalent of [$U$-$^{15}$N] Mfd-RID (blue). Both NMR titrations were performed in PBS buffer at 298 K.

RID, suggesting no apparent interaction between Mfd-RID and ssDNA (Supplementary Fig. 9a). Whereas, in the presence of dsDNA, we observed minute chemical shift changes in the Mfd-RID backbone amide resonances located mainly towards the amino- and carboxy-terminus of the protein (Supplementary Fig. 9b), indicative of unspecific interactions possibly due to a hydrophobic patch (e.g., L477, H478, I479) in the amino-terminus.

**UvrD-CTD and Mfd-RID RNAP binding sites are non-overlapping**. We next questioned whether these domains share the same RNAP-binding site or if they use different locations on RNAP for their interaction. In order to address this question, we designed a displacement/competition assay using NMR. Given the stronger binding affinity of UvrD-CTD to the RNAP in comparison to that of Mfd-RID (77 nM to 250 nM), we hypothesized that UvrD-CTD should effectively displace Mfd-RID from the Mfd-RID–RNAP complex if UvrD-CTD and Mfd-RID compete for the same binding site on the RNAP, otherwise addition of UvrD-CTD should not have any effect on the Mfd-RID–RNAP complex.

Hence, we pre-formed the complex between [$U$-$^{15}$N] labeled RID and RNAP showing the characteristic line-broadening of almost all the Mfd-RID backbone amide resonances in the 2D [$^{15}$N,$^{1}$H]-NMR spectrum (Fig. 6a). Upon addition of one molar equivalent of [$U$-$^{15}$N] labeled UvrD-CTD, we observed only very minute gain in the intensities of the few Mfd-RID backbone amide resonances located mainly in the loop regions, whereas majority of the backbone amide resonances did not show any intensity changes (Supplementary Fig. 10a). Even after incubation for eight hours, we did not observe any considerable gain in the intensity of the Mfd-RID resonances (Supplementary Fig. 10b, c) implying that there was no apparent disruption of the Mfd-RID-RNAP complex upon UvrD-CTD addition. Moreover, backbone amide resonances of the UvrD-CTD were also drastically line-broadened in line with UvrD-CTD-RNAP complex formation as our experimental set-up enabled us to observe both Mfd-RID and UvrD-CTD at the same time, indicating that the UvrD-CTD is able to interact with RNAP even in the presence of an intact

interaction with Mfd-RID. This observation was also confirmed by performing the experiment vice versa (Fig. 6b, Supplementary Fig. 10d, e). Together, this indicates that UvrD-CTD and Mfd-RID do not compete with each other for their respective interaction with RNAP and use two distinct binding sites.

To address the ensuing hypothesis that both proteins could simultaneously bind RNAP, we repeated the titrations with full-length UvrD and Mfd by pre-forming the equimolar complexes of [$U$-$^{2}$H$^{15}$N] Mfd–RNAP and [$U$-$^{2}$H$^{15}$N] UvrD–RNAP, characterized by the line-broadening of backbone amide resonances except for the flexible regions mainly observed at the center of the 2D [$^{15}$N,$^{1}$H]-NMR spectra (Supplementary Fig. 11a, b). Upon addition of one molar equivalent of [$U$-$^{2}$H$^{15}$N] labeled UvrD or Mfd, respectively, again we did not detect any apparent intensity increases of the backbone amide resonances indicating no disruption of the protein-RNAP complexes, clearly indicating that both full-length UvrD and Mfd are unable to dissociate each other from the RNAP complexes. To corroborate our analysis, we performed the size exclusion chromatography on the samples used for the NMR experiments (Supplementary Fig. 11c, d). We did not observe any apparent coelution for the RNAP-Mfd-UvrD ternary complexes, rather the pre-formed RNAP-Mfd and RNAP-UvrD complexes eluted distinctly, clearly showing that UvrD and Mfd are not able to bind RNAP simultaneously in vitro.

## Discussion

In this study, we used solution NMR spectroscopy to elucidate the structure of the extreme carboxy-terminal region of the UvrD helicase, implicated in RNA polymerase interactions, which was previously deemed to be disordered[16,39]. We show that UvrD$^{673-720}$ harbors a classical Tudor-domain-like fold, consisting of five highly bent anti-parallel β-strands folded into a barrel-type shape with a characteristic aromatic cage formed by four aromatic residues[40]. The first 29 residues of the amino-terminal region, bridging the Tudor-domain to the rest of the UvrD domains, are unstructured ensuring the flexible attachment of the CTD to the rest of the UvrD protein. Recent studies have highlighted the importance of such unstructured regions

governing allosteric and conformational changes playing a crucial role in the function of the connecting domains[41]. Thus, this amino-terminal region is highly likely a key for imparting high degrees of freedom to the Tudor domain fold, which might be essential for its proposed role as a freely accessible protein-protein binding hub. Remarkably, despite adopting a stable fold the UvrD-CTD shows extensive inherent fast dynamics of the aromatic cage forming residues (Fig. 2), providing a highly dynamic binding surface for the promiscuous interactions with a large diversity of binding partners.

UvrD's general DNA binding ability is crucial for many biological functions. Though crystallographic studies indicate that DNA binding is mainly driven by UvrD's 1B and 2B domains, the DNA binding contribution from the CTD remained elusive. The finding that UvrD-CTD recognizes both single-stranded and double-stranded DNA non-specifically is therefore perfectly in line with studies performed on other Tudor domains highlighting their emerging role in nucleic acid binding[35,36]. The common feature of this interaction is the contribution from the loops adjoining the β-strands mediating weak DNA binding. Despite the lack of high conservation between the DNA binding residues, the binding surface utilized for the DNA interaction by different Tudor domains remains similar[35].

Our solution NMR study agrees well and substantially extends the previous study reporting the crystal structure of the Tudor-domain of the carboxy-terminal region from the UvrD-homolog PcrA. Despite having high structural and sequence similarity between the Tudor-domains of the carboxy-terminal regions of these two proteins, we also notice subtle functional differences between these two proteins, like the lack of DNA binding capacity of PcrA-CTD[26]. The structural comparison suggests differences in terms of the aromatic cage between two proteins where the key aromatic residue W673 from PcrA-CTD has a different position with respect to the UvrD-CTD (Supplementary Figs. 3c and 12a). Further, we observe a different overall distribution of the electrostatic surface charges, (Supplementary Figs. 3b and 12b). Together these subtle alterations likely explain the differential DNA binding properties of these two related domains.

We also characterized the isolated RID (RNAP interaction domain) of Mfd, by solution NMR and compared its structure, RNAP-, and DNA-binding properties directly with UvrD-CTD. We observe the structural similarity in terms Tudor-domain fold for UvrD CTD and Mfd RID even though at the primary amino acid sequence level showing marked differences (Supplementary Fig. 12c). On the contrary, aromatic residues of the RID Tudor domain (five tyrosine residues) are not clustered together,

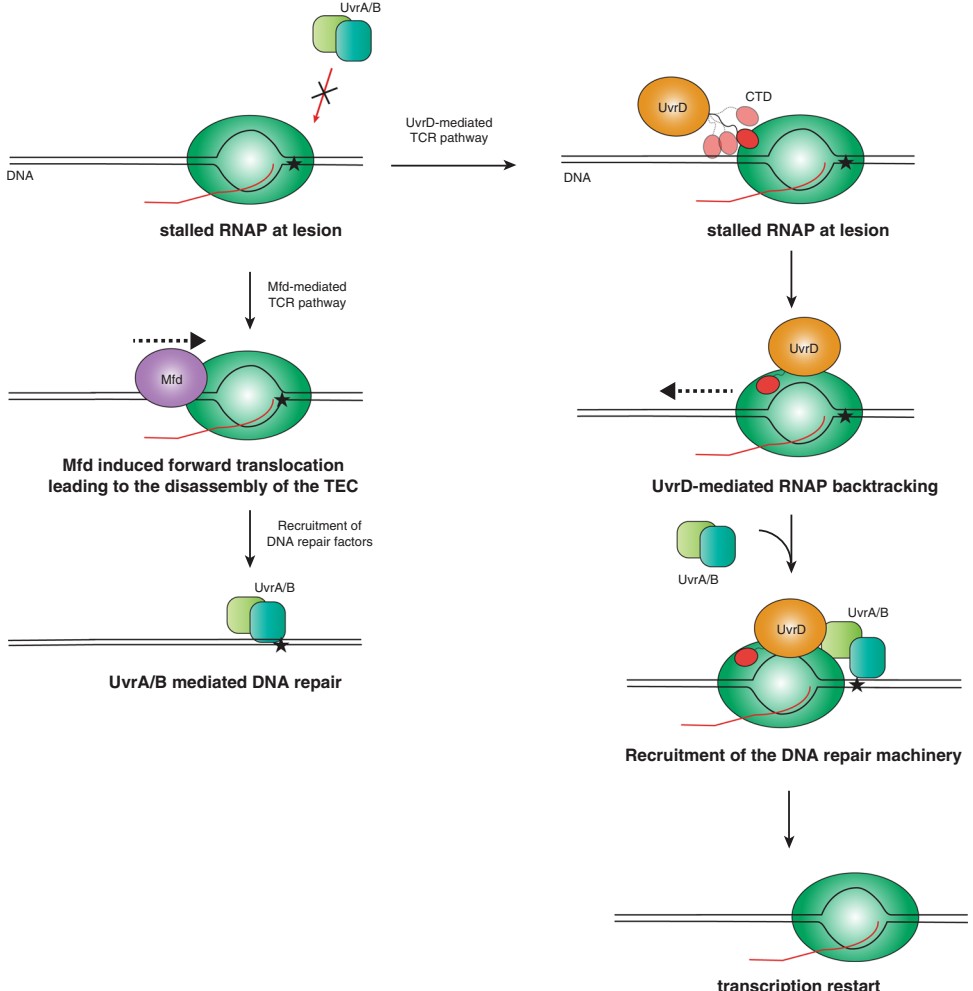

**Fig. 7 The proposed mechanistic role of UvrD-CTD.** Schematic diagram of the potential mechanism of actions of Mfd- and UvrD-mediated TCR pathway. Initial binding of UvrD to a stalled TEC via a direct UvrD-CTD–RNAP interaction could be governed by the flexibility of the CTD, leading to the formation of a stable UvrD-RNAP complex, enabling RNAP to backtrack and recruiting of the DNA repair machinery to the DNA lesion. In case a DNA lesion is encountered, the binding site could become available for the CTD enabling RNAP to enter the UvrD-mediated TCR pathway. In case the CTD is unable to bind because the RNAP binding site is occluded by other factors Mfd can induce RNAP forward translocation leading to the disassembly of the transcription bubble followed by recruitment of the DNA repair factors.

preventing the formation of aromatic cage or base stacking interactions crucial for DNA binding together with the absence of positive surface potential, explaining the lack of interaction between Mfd-RID and ssDNA. Furthermore, Mfd-RID does not show any enhanced internal backbone dynamics, which possibly can be attributed to its sole function to facilitate the Mfd–RNAP interaction.

Competitive binding assays revealed that UvrD-CTD and Mfd-RID do not compete with each other for RNAP binding but rather use distinct binding sites on RNAP in line with previous findings[2,4,8,38]. Full-length UvrD and Mfd also failed to dislodge each other from the respective RNAP complexes corroborating our observation deduced from the isolated RNAP binding domains. Nevertheless, both Mfd and UvrD full-length are also unable to dislodge RNAP complexes of their counterparts, implying that though UvrD and Mfd use different binding sites on RNAP for interactions via their respective RNAP binding domains, full-length proteins are unable to co-interact with RNAP possibly because of either auxiliary RNAP binding sites or of the steric hindrance posed by the other domains of respective proteins.

In summary, we show that although UvrD-CTD and Mfd-RID share the similar Tudor-domain-like fold, they are showing a marked difference in terms of DNA and RNAP binding properties and might follow different binding modes to carry out their respective TCR pathways. The UvrD-CTD might in this context be crucial for recruiting UvrD to stalled RNAPs to induce backtracking of the RNAP to recruit the DNA repair machinery (Fig. 7). Although the spatiotemporal details regarding the association of Mfd and UvrD with RNAP remain elusive, a concerted action of both proteins as initially hypothesized by Nudler and coworkers can possibly be ruled out based on our in vitro experiments[9]. Nevertheless, transcription regulation is a highly dynamic process with a large diversity of factors binding to RNAP at different phases[10,42,43]. In this context, both UvrD and Mfd have been reported to be able to remain associated with transcribing RNAP throughout large parts elongation process[44]. Future structural studies are required to decipher the detailed mechanistic details of the individual protein domains into the UvrD-facilitated TCR pathway.

## Methods
**Cloning**. The used UvrD constructs were subcloned from a pET28b-UvrD construct (a kind gift from E. Nudler) expressing tagless full-length UvrD. Further subcloning was performed by standard cloning techniques in a pET28b(+) modified vector backbone containing an uncleavable amino-terminal His$_6$-tag via the NcoI-XhoI restriction sites: Full-length (1–720), CTD (645–720). The UvrD-ΔCTD (1–664) construct was generated by inserting a stop codon by site-directed mutagenesis. The Mfd full-length (1–1148) construct was purchased from Genscript in a pET28b(+) vector encompassing NheI–EcoRI restriction sites. Mfd RID (472–547) and truncated Mfd RID (472–533) constructs were subcloned into a pET28b(+) vector and expressed as amino-terminal His$_6$-tag-Sumo-fusion proteins. All used plasmids and the sequences of the used primers are given in the Supplementary Material (Supplementary Tables 3, 4). UvrD constructs were transformed into chemically competent *E. coli* BL21(λDE3) pLysS cells, whereas Mfd constructs were transformed into chemically competent *E. coli* BL21 Star™ (λDE3) cells.

**Isotope labeling, protein expression, and purification**. Luria-Bertani (LB) or minimal M9 media supplemented with either ($^{15}$NH$_4$)Cl and *D*-($^{13}$C)-glucose were used for the expression of the protein samples in the unlabeled form or for uniformly labeled [*U*-$^{15}$N] or [*U*-$^{15}$N,$^{13}$C] proteins, respectively[45]. The deuterated [*U*-$^2$H,$^{15}$N]-UvrD and [*U*-$^2$H,$^{15}$N]-Mfd full length proteins were expressed in minimal medium supplemented with ($^{15}$NH$_4$)Cl in 99.8% D$_2$O. All isotopes were purchased from Merck.

Protein expression was induced by the addition of 1 mM isopropyl β-D-thiogalactoside (IPTG) at an OD$_{600}$ ~ 0.6 followed by 20 h of incubation at 25 °C. Following expression, bacterial cells were harvested by centrifugation at 4000 × *g* for 20 min at 4 °C. The ensuing cell pellet was resuspended in 50 ml of lysis buffer (50 mM Tris/HCl, pH 8.0, 500 mM NaCl, 5 mM Imidazole supplemented with one cOmplete, EDTA-free Protease Inhibitor Cocktail tablet (Roche), HL-SAN DNase I (ArticZymes) and 10 mM MgSO$_4$) per 10 g of wet cell-pellet weight. Cells were

disrupted by three passes through an Emulsiflex C3 (Avestin) homogenizer at 4 °C. Cell debris was separated using high-speed centrifugation at 19,000 × *g* for 45 min at 4 °C. The cleared lysate was loaded at least twice onto a manually packed Ni$^{2+}$–NTA (HisPur™ resin, Thermo Fisher Scientific) gravity column pre-equilibrated with lysis buffer. Non-specifically bound proteins were removed by passing 10 column volumes (CV) of lysis buffer as well as an additional washing step of lysis buffer supplemented with 25 mM Imidazole. Finally, bound proteins were eluted using elution buffer (lysis buffer + 250 mM Imidazole) for 5 CV. To inhibit eventual protease activity as well as to remove the contaminating divalent cations, 5 mM EDTA was added to the pooled elution fractions containing target proteins. Vivaspin 15 R centrifugal concentrators (Sartorius) with 3 kDa, 10 kDa, or 30 kDa MWCO (Molecular weight cut-off) were used depending upon the molecular weight of the respective protein to concentrate the elution fractions up to 1–2 ml, which were subsequently applied to a HiLoad 16/600 Superdex75 prep grade column (GE Healthcare) pre-equilibrated with 20 mM Tris/HCl, pH 8.0, 1 M NaCl, 2 mM DTT for subsequent purification.

For Sumo-tag containing constructs, overnight dialysis and subsequent Sumo-tag cleavage by human sumo protease (His-tagged SenP1; Addgene #16356)[46] were performed in 50 mM Tris/HCl, pH 8.0, 150 mM NaCl, 1 mM DTT containing buffer. A second Ni$^{2+}$–NTA gravity column purification step was performed to remove the cleaved Sumo-tag as well as SenP1. Flow-through fractions containing the protein of interest were subsequently used to perform size exclusion chromatography as outlined above.

**RNAP core enzyme expression and purification**. RNAP core enzyme was expressed from plasmid pIA900 (Addgene #104401;[47]) the expression and purification was performed as outlined by Svetlov and Artsimovitch[47] with an additional size exclusion chromatography step performed using a Superose6 10/300 GL column (GE Healthcare) pre-equilibrated with PBS Buffer at 4 °C instead of an ion exchange chromatography step. The presence of the intact RNAP core enzyme was confirmed by SDS-PAGE.

**DNA oligonucleotides**. Single-stranded DNA oligonucleotides (17mer: 5′-GCAG TGCTCGTTTTTTT-3′) and self-complementary oligonucleotides (35mer: 5′-CGA GCACTGCACTCGAGTGCAGTGCTCGTTGTTAT-3′) were purchased from Eurofins in a lyophilized form. They were subsequently dissolved in H$_2$O to a concentration of 2 mM and snap cooled on ice for 2 min before usage.

**SEC-MALS**. SEC-MALS experiments were performed using a Superdex Increase 200 10/300 GL column (GE Healthcare) on an Agilent 1260 HPLC Infinity II in PBS buffer at RT (~297 K). Protein elution was monitored by three detectors in series namely, an Agilent multi-wavelength absorbance detector (absorbance at 280 nm and 254 nm), a Wyatt miniDAWN TREOS multiangle light scattering (MALS) detector, and a Wyatt Optilab rEX differential refractive index (dRI) detector. The column was pre-equilibrated overnight in the running buffer to obtain stable baseline signals from the detectors before data collection. Molar mass, elution concentration, and mass distributions of the samples were calculated using the ASTRA 7.1.3 software (Wyatt Technology). A BSA solution (2–4 mg/ml), purchased from Sigma-Aldrich and directly used without further purification, was used to calibrate inter-detector delay volumes, band broadening corrections, and light-scattering detector normalization using standard protocols within ASTRA 7.1.3.

**NMR spectroscopy**. All NMR experiments for UvrD-CTD were recorded in NMR-buffer (20 mM potassium phosphate, 50 mM KCl, 5 mM DTT, pH 6.5 supplemented with 10% D$_2$O) at 310 K. For Mfd-RID, NMR experiments were recorded in PBS buffer, pH 7.4 supplemented with 10% D$_2$O at 298 K. NMR spectra were recorded on Bruker Avance III 700, 800 MHz spectrometers, equipped with either 5 mm QCI-F, 5 mm TXO, or 3 mm TCI cryoprobes, respectively, all running TopSpin3.5 (Bruker Biospin). NMR data processing was performed using TopSpin4.0.4 (Bruker Biospin), mddNMR2.4[48], and nmrPipe[49]. NMR spectral analysis was performed using NMRFAM-SPARKY[50]. For the sequence-specific backbone resonance assignment of the UvrD-CTD, the following experiments were performed: 2D [$^{15}$N,$^1$H]-TROSY-HSQC, 3D HNCA, HNCACB, HNCO, HNCACO and, CBCA(CO)NH triple-resonance experiments[51]. For Mfd-RID, BEST-type triple resonance experiments were used for the sequence-specific backbone resonance assignment[52]. $^1$H chemical shifts were directly referenced to DSS (4,4-dimethyl-4-silapentane-1-sulfonic acid) and for the $^{13}$C and $^{15}$N indirectly by standard methods. Chemical shift derived secondary structure elements were determined using sequence corrected random coil shifts generated by the POTENCI algorithm[53]. Noise reduction was achieved by treating the raw data with a 1-2-1 smoothening function for residues $(i − 1) − (i) − (i + 1)$ to highlight regular secondary structural elements as described before[54,55].

Aliphatic side-chain resonance assignment was performed based on 2D [$^{13}$C,$^1$H]-HMQC spectra with/without constant time version, 3D (H)CC(CO)NH, H(CC)(CO)NH, HCCH-TOCSY, $^{15}$N- and $^{13}$C-edited 3D-NOESY-HSQC experiments[51]. Aromatic side-chain resonance assignments were achieved using (Hβ)Cβ(CγCδ)Hδ and (Hβ)Cβ(CγCδCε)Hε [56], aromatic 2D [$^{13}$C,$^1$H]-constant time (CT)-HSQC as well as aromatic $^{13}$C-edited 3D-NOESY spectra.

Backbone [15]N relaxation experiments were recorded on a 700 MHz Bruker NMR spectrometer at 650 μM protein concentration at 310 K and 510 μM protein concentration at 298 K for UvrD-CTD and Mfd-RID, respectively. Steady-state heteronuclear 2D [15]N{[1]H}-NOE and TROSY-based NMR relaxation experiments were measured as described[57]. $R_{1\rho}$ data measurements were performed by recording delays of 0, 20, 40, 60, 80, 100, and 120 ms. The obtained $R_{1\rho}$ relaxation rate was then converted to the $R_2$ relaxation rate for each residue using the relation

$$R_1\rho = R_1\cos^2\theta + R_2\sin^2\theta \tag{1}$$

where $\theta = \tan^{-1}(v_1/\Delta v)$ and $\Delta v$ is the offset of the rf field to the resonance[58]. Additionally, the TROSY ($R_{2\beta}$) and anti-TROSY lines ($R_{2\alpha}$) where recorded by relaxation delay points such as 4, 10, 20, 30, 40 ms as well as 0.4, 2, 4, 8, 16 ms, respectively. The transverse cross-correlated relaxation rate $\eta_{xy}$ has been extracted from the difference between the $R_{2\beta}$ and $R_{2\alpha}$ rates. $R_1$ measurements were performed with delays of 400, 600, 800, 1200, 1600, 1800, 2400, and 3200 ms. NMRFAM-SPARKY was used for analyzing the relaxation data and further analysis of the relaxation data was performed by in-house written scripts in Matlab (MathWorks) and with the TENSOR2 program[59] used via NMRbox[60] using an axially symmetric diffusion tensor.

**NMR titrations.** Protein–protein and protein–nucleic-acid titrations were performed by recording 2D [[15]N,[1]H]-SOFAST-HMQC or 2D [[15]N,[1]H]-TROSY-HSQC experiments in the presence and the absence of the respective ligands in NMR buffer or PBS buffer at 298 K. The chemical shift perturbation of the amide moiety in the presence of ligand were calculated by using the following equation

$$\Delta\delta(HN) = \sqrt{(\Delta\delta^1H)^2 + (\Delta\delta^{15}N/5)^2} \tag{2}$$

**Structure calculation.** Solution NMR structure calculations were performed using distance and torsion angle restraints in CYANA version 3.98.12[61]. Distance restraints were obtained using [15]N- and [13]C-edited 3D-NOESY-HSQCs (for both aliphatic as well as aromatic protons) recorded in protonated buffers with 120 ms mixing time. NMRFAM-SPARKY was used for peak picking and spectra alignments. The CYANA combined automated NOESY cross peak assignment and structure calculation protocol was used for obtaining distance restraints[62]. TALOS+ was used for generating torsion angle restraints[63]. The resulting best 20 structural models with the lowest energy from CYANA were used for subsequent water refinement by ARIA CNS[64]. The refined structures were validated using the wwPDB web server (https://validate.wwpdb.org). The 10 conformers with the lowest distance violations were chosen for the final structural bundle. The structural statistics are given in Table 1. Ramachandran plot analysis was performed by using RAMPAGE (http://www-cryst.bioc.cam.ac.uk/rampage/). The percentage of residues in the most favored, additionally allowed and disallowed regions are 83.9, 13.7 and 2.4 (UvrD-CTD) and 90.1, 9,5 and 0.0 (Mfd-RID), respectively. Structure figures were prepared with the open-source version of PyMOL (1.8.x) Schrödinger, LLC. Electrostatic surface representations were generated using the APBS plugin[65].

**Bio-layer interferometry (BLI).** BLI experiments were performed on an Octet RED96 system (Fortébio) at 303 K. The respective ligands were biotinylated using the biotinylation kit EZ-Link NHS-PEG4-Biotin (Thermo Fisher Scientific). The biotin label was freshly resolved in $H_2O$, directly added to the protein solution in a final molar ratio of 1:1 in PBS buffer supplemented with 2.5 mM MgCl$_2$ and 2 mM DTT followed by gentle mixing at room temperature for 45 min. The used reaction conditions (phosphate buffer pH 7.4) favored the preferential labeling of the N-terminal α-amino group of proteins[66]. Unreacted biotin was removed using Zeba Spin Desalting Columns (7 MWCO, Thermo Fisher Scientific). Biotin-labeled proteins were immobilized on the streptavidin (SA) biosensors (Fortébio) and the biosensors were subsequently blocked with EZ-Link Biocytin (Thermo Fisher Scientific). Analytes were diluted and applied in a dose-dependent manner to the biosensors immobilized with the biotinylated ligand. Bovine serum albumin (BSA) powder (Sigma-Aldrich) was added to a final concentration of 0.1% to avoid non-specific interactions. Parallel experiments were performed for reference sensors with no analyte bound and the signals were subsequently subtracted during data analysis. The association and dissociation periods were both set to 1000 s. Data measurements and analysis were performed by using the Data acquisition 10.0 and the Data analysis HT 10.0 (Fortébio) software, respectively.

## Data availability

The source data underlying main figures are provided as Supplementary Data 1 file. The solution NMR-derived structures of the UvrD-CTD, Mfd-RID have been deposited in the PDB under entries 6YI2 and 6YHZ, all the sequence-specific NMR resonance assignments in the BMRB with accession codes 50218 and 50219, respectively. All other relevant data are available from the corresponding author upon reasonable request.

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

## Acknowledgements

B.M.B. gratefully acknowledges funding from the Swedish Research Council (Starting Grant 2016-04721) and the Knut och Alice Wallenberg Foundation through a Wallenberg Academy Fellowship (2016.0163) as well as through the Wallenberg Centre for Molecular and Translational Medicine, University of Gothenburg, Sweden. Members of the Burmann Lab are acknowledged for helpful discussions. The Swedish NMR Centre of the University of Gothenburg is acknowledged for spectrometer time. This study made use of NMRbox: National Center for Biomolecular NMR Data Processing and Analysis, a Biomedical Technology Research Resource (BTRR), which is supported by NIH grant P41GM111135 (NIGMS).

## Author contributions

B.M.B. conceived the study and designed the experiments together with A.A.K. A.A.K. performed all experimental work. A.A.K. and B.M.B. analyzed and discussed the data. A.A.K. and B.M.B. jointly wrote the paper.

## Funding

## Competing interests

The authors declare no competing interests.
