## [Peer Review File · Communications Biology]

Reviewers' comments:

Reviewer #1 (Remarks to the Author):

Manuscript by Kawale and Burmann fills an important structural and biochemical gap in understanding of the cellular function of UvrD, a crucial component in DNA damage surveillance and repair. The authors report the structure of UvrD C-terminus, which was unresolved or apparently disordered in past crystallographic models. By using NMR spectroscopy, the authors were able to solve the structure of the C-terminal Tudor-like domain, refractory to the conventional X-ray crystallography. The manuscript under review also provide important insights into the role this domain plays in interactions with RNA polymerase, single- and double-stranded DNA during transcription-coupled repair (TCR), and sheds light on the interplay between UvrD and another TCR factor, Mfd/TCRF.

The report is technically sound, with well-planned and executed experiments, clearly written and supplies sufficient details for reproducibility. The solution NMR pipeline is benchmarked against published structural model of the Tudor domain present in Mfd. In addition to insights into mechanism of UvrD-dependent TCR, the manuscript can serve as a guide to using NMR as complimentary tool in structural analysis of multi-domain proteins.

I recommend the manuscript for publication with minor modifications to address my specific comment below:

Throughout the report the authors used BLI (aka SPR) to measure dissociation constants of UvrD-CTD, full-length UvrD, and its truncated version in complexes with RNAP, ss- and dsDNA, etc. Given the fact that the authors used NHS-ester-mediated biotinylation of the UvrD derivatives, one should be more cautious comparing the K_d values thus obtained. One of the well-established caveats of BLI/SPR is the impact of the immobilization approach, and non-random/preferred orientation of the immobilized solute. UvrD-CTD has only 3 lysine residues that can react with NHS-biotin, making the orientation of this domain on the surface substantially less random than that of full-length or truncated UvrD. While this caveat doesn't invalidate the analysis, the quantitative differences between full-length UvrD (720 amino acids, 24 lysines) and UvrD-CTD (75 amino acids, 3 lysines) should be viewed with caution. A sentence (or to this effect) would be a prudent addition to the report under review.

Reviewer #2 (Remarks to the Author):

In the article by Kawale & Burmann the authors describe the solution structure of the CTD domain from the bacterial UvrD helicase which is similar to the TUDOR domain fold, and characterize its binding to DNA and to RNA polymerase. They also report the C-terminal domain structure of the transcription-repair-coupling factor Mfd and its interaction with DNA and RNA polymerase. They lead to a model in which the C-terminal domains of UvrD and Mfd bind to separate sites on RNA polymerase.

The two separate structure calculations appear to have been carefully performed and the resulting structural statistics support high quality models. The structural details have also been well described and the support NMR data on properties such as backbone dynamics provide a meaningful analysis.

My main concerns deal mostly with the final experiments that lead to the model of separate binding regions on RNA polymerase. This conclusion derives mainly from two experiments - a competition binding assay by NMR spectroscopy, and a second that relies on size exclusion chromatography. In the NMR competition assay, for example in Figure 6a, it is shown that ¹⁵N-labelled seems to bind to RNAP already in complex with the RID domain of Mfd. It would be good to add more caveats of these results, such as possible roled in dynamics during the formation of

the complexes - such as do the conclusions assume that the spectra follow complex formation at a slow NMR timescale. How would an intermediate timescale interaction behave with respect to the binding model? Also why do some of the red peaks (^{15}N -UvrD-CTD) overlay perfectly with the unbound Mfd-RID peaks. Does this in fact show that some Mfd-RID is displaced? In the size exclusion chromatography I may have misunderstood the experiment approach, but it is stated that only the pre-formed RNAP-Mfd or RNAP-UvrD complexes were observed which supported the model of non-overlapping binding regions on RNAP. Doesn't this in fact show that the opposite is correct? If the binding sites on RNAP were separate it would be expected that the authors would detect at least some complex in which both UvrD and Mfd were bound to RNAP. Some additional analysis and discussion are required in this section to explain how the model is supported. At present the model is not clearly supported by the reported data.

Other concerns:

More discussion is needed to explain possible reasons for the significant reduction in affinity to RNAP displayed by full-length UvrD as compared to the isolated CTD domain. This is especially true since the removal of the CTD in the deletion construct also only decreases affinity by a factor of 2.

The DNA ligand used for dsDNA appears to have seven nucleotides on each side that are in fact single-stranded DNA. Would this affect the conclusions of binding for ssDNA vs dsDNA?

It might be useful to rearrange the manuscript so that the section on UvrD CTD structure and NMR characterization is directly followed by the binding studies with UvrD. And then after these sections then put the structure determination of the Mfd RID along with a clear link to explain why this second domain is being investigated. Afterward would be the remaining binding studies with Mfd and the final section on how the two proteins, UvrD and Mfd may interact with RNAP.

Minor points :

- an additional sentence would help to understand how the boundaries of the UvrD CTD were chosen (Page 5, start of the results)

- on Page 5 it is stated that the value for the rotational correlation time was less than expected, possibly due to the presence of flexible portions of the construct. I would expect that the flexible tail would in general increase the effective size of the molecule resulting in a larger than expected correlation time.

Response to Reviewers' Comments

We thank the reviewers for providing helpful critical comments on our manuscript draft. As detailed below, we have addressed the concerns, including the additional analysis and modifications to the manuscript draft to our best ability.

Reviewers' comments:

Reviewer #1 (Remarks to the Author):

Manuscript by Kawale and Burmann fills an important structural and biochemical gap in understanding of the cellular function of UvrD, a crucial component in DNA damage surveillance and repair. The authors report the structure of UvrD C-terminus, which was unresolved or apparently disordered in past crystallographic models. By using NMR spectroscopy, the authors were able to solve the structure of the C-terminal Tudor-like domain, refractory to the conventional X-ray crystallography. The manuscript under review also provide important insights into the role this domain plays in interactions with RNA polymerase, single- and double-stranded DNA during transcription-coupled repair (TCR), and sheds light on the interplay between UvrD and another TCR factor, Mfd/TCRF. The report is technically sound, with well-planned and executed experiments, clearly written and supplies sufficient details for reproducibility. The solution NMR pipeline is benchmarked against published structural model of the Tudor domain present in Mfd. In addition to insights into mechanism of UvrD-dependent TCR, the manuscript can serve as a guide to using NMR as complimentary tool in structural analysis of multi-domain proteins.

I recommend the manuscript for publication with minor modifications to address my specific comment below:

We gratefully thank this reviewer for his appreciation of our work.

Throughout the report the authors used BLI (aka SPR) to measure dissociation constants of UvrD-CTD, full-length UvrD, and its truncated version in complexes with RNAP, ss- and dsDNA, etc. Given the fact that the authors used NHS-ester-mediated biotinylation of the UvrD derivatives, one should be more cautious comparing the K_d values thus obtained.

One of the well-established caveats of BLI/SPR is the impact of the immobilization approach, and non-random/preferred orientation of the immobilized solute. UvrD-CTD has only 3 lysine residues that can react with NHS-biotin, making the orientation of this domain on the surface substantially less random than that of full-length or truncated UvrD. While this caveat doesn't invalidate the analysis, the quantitative differences between full-length UvrD (720 amino acids, 24 lysines) and UvrD-CTD (75 amino acids, 3 lysines) should be viewed with caution. A sentence (or to this effect) would be a prudent addition to the report under review.

Although the mentioned caveat is valid point, we believe to already taken the issue raised in to account. On the one hand we have used for the labeling a 1:1 molar ratio of biotin to the protein, to reduce the chance of multiple biotinylation. On the other hand, the chosen

conditions for the biotinylation reaction (PBS Buffer pH=7.4), would favor labeling of the N-terminal α -amino group of proteins, as described by Sélo, *et al.*, J. Immunol. Methods, 1996 that lower reaction pH ensures that the lysine amines are very rarely in the reactive unprotonated state as the pKa of the α -amino group (pKa = 8.9) is considerably lower than that of the ϵ -amino group of lysine (pKa = 10.5). Preferred N-terminal biotinylation lowers the risk of mentioned caveat. To also make this point clear within the methods section we added the following sentence including the mentioned reference:

The used reaction conditions (phosphate buffer pH 7.4) favored the preferential labeling of the N-terminal α -amino group of proteins ⁶⁵.

Reviewer #2 (Remarks to the Author):

In the article by Kawale & Burmann the authors describe the solution structure of the CTD domain from the bacterial UvrD helicase which is similar to the TUDOR domain fold, and characterize its binding to DNA and to RNA polymerase. They also report the C-terminal domain structure of the transcription-repair-coupling factor Mfd and its interaction with DNA and RNA polymerase. They lead to a model in which the C-terminal domains of UvrD and Mfd bind to separate sites on RNA polymerase.

The two separate structure calculations appear to have been carefully performed and the resulting structural statistics support high quality models. The structural details have also been well described and the support NMR data on properties such as backbone dynamics provide a meaningful analysis.

My main concerns deal mostly with the final experiments that lead to the model of separate binding regions on RNA polymerase. This conclusion derives mainly from two experiments - a competition binding assay by NMR spectroscopy, and a second that relies on size exclusion chromatography. In the NMR competition assay, for example in Figure 6a, it is shown that 15n-labelled (UvrD-CTD) seems to bind to RNAP already in complex with the RID domain of Mfd. It would be good to add more caveats of these results, such as possible roles in dynamics during the formation of the complexes - such as do the conclusions assume that the spectra follow complex formation at a slow NMR timescale. How would an intermediate timescale interaction behave with respect to the binding model? Also why do some of the red peaks (15N-UvrD-TCD) overlay perfectly with the unbound Mfd-RID peaks. Does this in fact show that some Mfd-RID is displaced?

As suggested by the reviewer, we have added an additional supplementary figure 10, showing each spectrum of the displacement assay side by side for better clarity and annotated the Mfd-RID resonances showing apparent marginal gain in the intensity after addition of UvrD-CTD to the Mfd-RID-RNAP complex and rephrased the paragraph describing the results for better understanding as follows:

Upon addition of one molar equivalent of [U - ^{15}N] labeled UvrD-CTD, we observed only very minute gain in the intensities of the few Mfd-RID backbone amide resonances located mainly in the loop regions, whereas majority of the backbone amide resonances did not show any

intensity changes (Supplementary Fig. 10 a). Even after incubation for eight hours, we did not observe any significant gain in the intensity of the Mfd-RID resonances (Supplementary Fig. 10 b, c) implying that there was no apparent disruption of the Mfd-RID-RNAP complex upon UvrD-CTD addition.

The analysis clearly shows that the Mfd-RID resonances gaining marginal intensity are mainly located in flexible parts of the protein and the intensity changes ~5% at maximum. Although any conformational and dynamical changes related to Mfd-RID and UvrD-CTD binding to RNAP are not known and cannot be resolved with our assay, given the stronger affinity of UvrD-CTD compared to Mfd-RID, a global recovery of the Mfd-RID resonances would be expected in the case both proteins would compete for the same binding site on RNAP. The fact that we did not observe further changes in the RID resonances even after ~8 hours of incubation at 298 K of Mfd-RID-RNAP complex in the presence of equimolar UvrD-CTD, implies that UvrD-CTD cannot dislodge the Mfd-RID from RNAP.

Though, protein-protein binding involving at the intermediate exchange regime, do undergo line broadening of the resonances, the above assumption that increase in the free form of the protein would lead globally to a decrease in the observed line-width of Mfd-RID resonances would remain still valid and thereby, we think that it does not invalidate our conclusion.

In the size exclusion chromatography I may have misunderstood the experiment approach, but it is stated that only the pre-formed RNAP-Mfd or RNAP-UvrD complexes were observed which supported the model of non-overlapping binding regions on RNAP. Doesn't this in fact show that the opposite is correct? If the binding sites on RNAP were separate it would be expected that the authors would detect at least some complex in which both UvrD and Mfd were bound to RNAP. Some additional analysis and discussion are required in this section to explain how the model is supported. At present the model is not clearly supported by the reported data.

Our results suggest that RNAP binding domains from UvrD and Mfd (UvrD-CTD and Mfd-RID, respectively) are able to bind RNAP simultaneously opening in general the possibility that UvrD and Mfd might interact simultaneously with RNAP. The results for full-length UvrD and Mfd proteins as seen by the lack of co-elution of Mfd-UvrD-RNAP ternary complexes in size-exclusion chromatography experiments on the other hand clearly show that the full-length proteins impair the binding of the respective other protein to RNAP. Based on the fact that we used initially the respective main RNAP binding domains leads to the following conclusions: (1) The main RNAP interaction sites are non overlapping. (2) The full-length proteins disable simultaneous binding either through auxiliary binding sites that might overlap or based on their large size steric hindrance. As at the moment the exact details of the respective UvrD-RNAP and Mfd-RNAP complexes are not known a more detailed interpretation is not possible. To address this issue, we clarified this aspect in the text as follows:

Nevertheless, both Mfd and UvrD full length are also unable to dislodge RNAP complexes of their counterparts, implying that though UvrD and Mfd use different binding sites on RNAP for interactions via their respective RNAP binding domains, full-length proteins are unable to co-

interact with RNAP possibly because of either auxiliary RNAP binding sites or of the steric hindrance posed by the other domains of respective proteins.

Other concerns:

More discussion is needed to explain possible reasons for the significant reduction in affinity to RNAP displayed by full-length UvrD as compared to the isolated CTD domain. This is especially true since the removal of the CTD in the deletion construct also only decreases affinity by a factor of 2.

We have updated results accordingly the following:

... weaker than the binding affinity of the isolated UvrD-CTD towards RNAP possibly suggesting possible steric hindrance imposed by the rest of the UvrD domains. Although the possibility of large conformational changes in the UvrD domain arrangement upon RNAP binding cannot be ruled out.

The DNA ligand used for dsDNA appears to have seven nucleotides on each side that are in fact single-stranded DNA. Would this affect the conclusions of binding for ssDNA vs dsDNA?

We appreciate the reviewer's comment as it outlines a valid point. The DNA oligonucleotides were chosen to according to the DNA-bound crystal structures reported for UvrD (Lee and Wang, Cell, 2006.) and the self-complementary oligonucleotides first 28 bp form dsDNA with 7 nucleotide 3' overhangs.

We have corrected the description of the DNA. As we have also performed DNA binding studies on ssDNA, the additional binding contributions might stem from the dsDNA part resulting in the increase the binding affinity, therefore the presence of the ss overhang might not any influence on the conclusions. The changed text reads now as follows:

Hence, in order to investigate the DNA binding properties of UvrD-CTD, we performed NMR titrations with a 17mer single-stranded DNA (ssDNA) and a self-complementary 35mer where the first 28 nucleotides form double-stranded DNA (dsDNA) with 3' overhangs comprising 7 nucleotides, used in structural studies before ¹⁵.

It might be useful to rearrange the manuscript so that the section on UvrD CTD structure and NMR characterization is directly followed by the binding studies with UvrD. And then after these section then put the structure determination of the Mfd RID along with a clear link to explain why this second domain is being investigated. Afterward would be the remaining binding studies with Mfd and the final section on how the two proteins, UvrD and Mfd may interact with RNAP.

Although the reviewer's thoughtful suggestion of possibly rearranging the manuscript is highly appreciated, we believe that the current set-up with the direct head-to-head comparison of CTD and RID dynamics followed by the differences observed for them in the binding studies is more suitable.

Minor points :

- an additional sentence would help to understand how the boundaries of the UvrD CTD were chosen (Page 5, start of the results)

The following sentence has been added clarifying the choice:

*This domain boundary was chosen as this region was missing from previously reported UvrD structures ^{15,16} and was derived from the structural studies observed for the *G. stearothermophilus* PcrA-CTD ²⁵.*

- on Page 5 it is stated that the value for the rotational correlation time was less than expected, possibly due to the presence of flexible portions of the construct. I would expect that the flexible tail would in general increase the effective size of the molecule resulting in a larger than expected correlation time.

Thank you very much for outlining this possible misunderstanding. The paragraph has been clarified. It's true that the presence of the flexible region will increase the effective size of the molecule resulting in the global increase in the correlation time. The NMR based determination of Rotational correlation time spherical approximation is used to derive the rotational correlation time. The use of simple 1D [¹⁵N-¹H]-TRACT approach has caveat which results in the shorter τ_c value than the expected effective rotational correlation time for an equivalent sphere if protein under observation has extensive flexible regions which is described in the Lee *et al.*, JMR, 2006.

Although the monomeric state is also reflected by the estimation of the effective rotational correlation time (τ_c) with 3.63 ns, the obtained value is about 30% lower than expected for a protein of this size (Fig. 1c). This significantly shorter than expected τ_c value is indicative of the presence of a highly flexible polypeptide segment dominating the analysis in the used approach ²⁶.

REVIEWERS' COMMENTS:

Reviewer #1:

Remarks to the Author:

The changes made by the authors in the revised manuscript have fully addressed my concerns. A new minor request is to cite PMID: 27199428. This work provides an independent in vivo evidence supporting the role of RNAP backtracking and UvrD in TCR.

Reviewer #2:

Remarks to the Author:

The changes made to the manuscript have addressed all of my concerns.

Reviewer #1 (Remarks to the Author):

The changes made by the authors in the revised manuscript have fully addressed my concerns. A new minor request is to cite PMID: 27199428. This work provides an independent in vivo evidence supporting the role of RNAP backtracking and UvrD in TCR.

We appreciate the support of the reviewer.

We have added the requested reference to the MS.

Reviewer #2 (Remarks to the Author):

The changes made to the manuscript have addressed all of my concerns.

We are glad to remove the concerns of this reviewer.